**www.cambridge.org/qrd**

## Perspective

RNA folding; enhanced sampling simulations; path sampling simulations; RNA coarse-grained models; energy landscapes

**Author for correspondence:**
*Samuela Pasquali,
E-mail: samuela.pasquali@u-paris.fr

# Computer-aided comprehensive explorations of RNA structural polymorphism through complementary simulation methods

Konstantin Röder[1] ![ORCID], Guillaume Stirnemann[2] ![ORCID], Pietro Faccioli[3] and Samuela Pasquali[4,5]* ![ORCID]

[1]Yusuf Hamied Department of Chemistry, University of Cambridge, Cambridge, UK; [2]CNRS Laboratoire de Biochimie Théorique, Institut de Biologie Physico-Chimique, PSL University, Université de Paris, 13 rue Pierre et Marie Curie, Paris 75005, France; [3]Department of Physics, University of Trento and INFN-TIFPA, Trento, Italy; [4]Laboratoire CiTCoM, CNRS UMR 8038, Université Paris Cité, 4 avenue de l'Observatoire, Paris 75006, France and [5]Laboratoire BFA, CNRS UMR 8251, Université Paris Cité, 35 rue Hélène Brion, Paris 75013, France

## Abstract

While RNA folding was originally seen as a simple problem to solve, it has been shown that the promiscuous interactions of the nucleobases result in structural polymorphism, with several competing structures generally observed for non-coding RNA. This inherent complexity limits our understanding of these molecules from experiments alone, and computational methods are commonly used to study RNA. Here, we discuss three advanced sampling schemes, namely Hamiltonian-replica exchange molecular dynamics (MD), ratchet-and-pawl MD and discrete path sampling, as well as the HiRE-RNA coarse-graining scheme, and highlight how these approaches are complementary with reference to recent case studies. While all computational methods have their shortcomings, the plurality of simulation methods leads to a better understanding of experimental findings and can inform and guide experimental work on RNA polymorphism.

## The complexity of RNA folding

After the seminal experiments showing the hierarchical folding of RNA, RNA folding was thought to be an easier problem to solve than protein folding (Tinoco and Bustamante, 1999; Li *et al.*, 2008). With an alphabet composed of only four letters, and with key interactions leading to the observed secondary structure dictated by canonical base pairing (G with C and A with T/U), what remained to be solved was 'only' a combinatorial problem of finding the best pairing scheme for a given sequence.

About two decades later, we know that the problem is much more complex. Even searching for the optimal secondary structure remains a challenge as exhaustive sampling of all relevant conformations is unfeasible for most systems of biological interest, even though the advent of machine learning and the extensive use of chemical probing data are contributing to making the problem more tractable (Lorenz *et al.*, 2016; Zhao *et al.*, 2021). A common feature in complex RNA architectures is pseudoknots –non-nested arrangements of base pairs. Traditional secondary structure prediction algorithms do not treat these structures well and combining these approaches with machine learning has led to some progress (Wang *et al.*, 2019; Sato and Kato, 2022). The situation is even more complex considering that canonical base pairing, even though dominant, is not the only form of base pairing. The multiple hydrogen bond donor and acceptor sites of the nucleobases allow for a multitude of base pairs, which have been reported experimentally. Around 150 non-canonical base pairs have been found and classified in terms of interaction 'edges' (Watson-Crick, Hoogsteen and Sugar) (Leontis and Westhof, 2001; Stombaugh *et al.*, 2009). The full list can be found in the RNA Basepair Catalog of the Nucleic Acids Databank.

As it is the case in general for heteropolymers, a smaller alphabet results in an increase of frustration of the conformational space accessible to the molecule. In the case of RNA, the alphabet composed of only four different nucleobases, further complicated by the multitude of possible base pairs, results in a folding process possibly more complex to predict than for proteins (Ferreiro *et al.*, 2014). The observation that proteins fold reliably and fast into their native confirmation has been explained by the principle of minimal frustration (Bryngelson and Wolynes, 1987). Every sequence defines interactions between different parts of the molecule. The more of these are formed, the lower the frustration and the more stable the resulting structure. The native state exhibits a conformation that fulfils all packing requirements, that is the system shows minimal frustration. Minimal frustration is linked to the topography of the energy landscape (EL), and in the case of globular proteins a single funnel anchored around the

native fold is observed (Leopold *et al.*, 1992; Bryngelson *et al.*, 1995).[1] As a result, the number of native contacts observed is a good proxy for the progress of the highly cooperative folding of proteins.

In contrast, RNA is characterised by the existence of several stable structural ensembles with different secondary structures, and many of these systems are highly dynamic (Brillet *et al.*, 2020). The number of alternative contacts in RNA leads to large frustration and disorder, as the sequence allows for multiple competing inter-actions. This higher frustration has been highlighted both by experiments (Burge *et al.*, 2006; Garst *et al.*, 2011; Martinez-Zapien *et al.*, 2017; Kolesnikova and Curtis, 2019; Lightfoot *et al.*, 2019; Saldi *et al.*, 2021; Yu *et al.*, 2021) and by simulations (Denesyuk and Thirumalai, 2011; Cragnolini *et al.*, 2015; Šponer *et al.*, 2018; Röder *et al.*, 2020; Schlick *et al.*, 2021; Rissone *et al.*, 2022; Röder *et al.*, 2022; Yan *et al.*, 2022), and its main manifestation is structural polymorphism. Within these distinct structures, there must not necessarily be a distinct global minimum, and therefore a native state does not necessarily exist, as has been noticed by others (Vicens and Kieft, 2022).

Therefore, in our opinion, an ensemble approach should be chosen when talking about RNA. The relative population of these structural ensembles depends on experimental conditions, as observed for riboswitches and several other non-coding regulatory RNAs (Halvorsen *et al.*, 2010; Fay *et al.*, 2017; Kolesnikova and Curtis, 2019; Brillet *et al.*, 2020). Post-transcriptional modifications and single-point mutations also can shift the equilibrium between the alternative structures (Liu *et al.*, 2017; Martinez-Zapien *et al.*, 2017; Schlick *et al.*, 2021; Röder *et al.*, 2022). Finally, many RNAs interact with proteins and these interactions often lead to changes to the observed fold (Jaeger *et al.*, 2009). Which structure is detected in experiments therefore depends on the details of the experiment itself, and at times more than one structure is detected in the same experiment (Martinez-Zapien *et al.*, 2017).

Given this plurality of possible structures, simulations cannot be limited to the prediction of a single structure (which is what is achieved by most bioinformatic approaches), and focus must shift to a global view, which centres around the molecular EL. All information about the structures, their energy and interconversion pathways between them can be calculated from knowledge of the EL. Insight can also be obtained on the influence of external factors such as ionic conditions, pH, temperature, presence of ligands and chemical changes in the sequence by considering the EL. Any experiment or simulation probes the EL directly or indirectly. Various methods do so in different ways, and often the EL is not directly mapped.

The most common simulation method is molecular dynamics (MD) simulations. However, due to the broken ergodicity exhibited biomolecular ELs (Wales and Salamon, 2014) there are many practical difficulties. In brief, the structural ensembles are separated by high barriers, making transitions between them rare events. This kinetic partition between different regions will make observation of transitions in standard MD simulations very unlikely. As a result, so-called enhanced sampling approaches have been developed, which, for example, include path sampling methods.

Here, we present our perspective on how simulations can be used to gather information on RNA ELs and structural polymorph-ism. There are two approaches commonly employed. The first

option is the use of enhanced sampling methods (Mlýnský and Bussi, 2018), and here we briefly present three of these, namely Hamiltonian-replica exchange [H-REX, Replica-Exchange with Solute Tempering (REST2)] simulations (Wang *et al.*, 2011), dis-crete path sampling (DPS) (Wales, 2002, 2004) and a variationally optimised ratchet-and-pawl MD (rMD) simulation scheme (Tiana and Camilloni, 2012) called Bias Functional (BF) approach (Beccara *et al.*, 2015). The second option is to smooth the EL through coarse graining (CG) (Papoian, 2018). A pictorial illustra-tion on how each of these methods samples the EL is given in Fig. 1. By considering several examples, we show that these approaches are complementary, and that the best results are obtained when com-bining multiple simulation methods.

## An overview of the simulation methods

### *H-REX simulations*

Despite the increased time scales that can be probed with unper-turbed MD simulations – now routinely on the order of $\mu$s – the relevant conformational motions cannot be sampled as the associated time scales still exceed computational feasibility, prompting interest for enhanced sampling strategies that have been developed and widely applied to biomolecules, including RNA (see, e.g., Mlýnský and Bussi, 2018 for a recent review).

One way to improve sampling in an unguided way (i.e. without assuming or imposing predetermined collective variables (CVs) along which the transitions will occur) is through the use of replica exchange MD simulations (Sugita and Okamoto, 1999). Multiple copies of the system are simulated at different temperatures, increasing the accessible time scales. However, this approach is very sensitive to the overlap between the replicas, which depends on the number of degrees of freedom, and at the moment it is hardly applicable at an all-atom resolution for nucleic acids exceeding a handful of residues in explicit solvent.

This problem may be overcome by using H-REX simulation schemes. In particular, we used the REST2 strategy (Wang *et al.*, 2011), where all replicas evolve at the same physical temperature, but they can exchange their Hamiltonian with a scaled potential energy for the biomolecule (Fig. 1a), decreasing the number of degrees of freedom. As a result, fewer replicas are required, and sampling is enhanced. For example, for proteins containing 100–200 residues, one to two dozen replicas were shown to lead to satisfactory exchange probabilities (Stirnemann and Sterpone, 2017; Maffucci *et al.*, 2020a, 2020b). However, this technique has mostly been applied to short oligonucleotides, and in particular to the sampling of tetraloops conformational space (Kührová *et al.*, 2016; Bottaro *et al.*, 2020; Mlýnský *et al.*, 2022). While a recent work pointed to limitations in the ability of such an approach to actually fold even short RNAs (Mlýnský *et al.*, 2022), REST2 remains a very attractive strategy to ease and to accelerate conformational sam-pling, which eventually enables to escape the kinetic traps in which brute-force simulations may be stuck for long times.

In this short perspective, we exclusively focus on REST2, which we have applied to an RNA much larger than these tetraloops (Röder *et al.*, 2020), but other applications to reasonably large biomolecules are mostly limited to DNAs and proteins. For these applications, recent success of REST2 in identifying important conformations that were not revealed by long brute-force MD (Stirnemann and Sterpone, 2017; Maffucci *et al.*, 2020a, 2020b; Gillet *et al.*, 2021) offers promising perspectives for the RNA field. However, it should be noted that when employed with atomistic

---

[1]This description extends to proteins that exhibit more than one structural ensemble, and which have a multifunnel energy landscape. Such landscapes are also governed by the principle of minimal frustration (Röder and Wales, 2018).

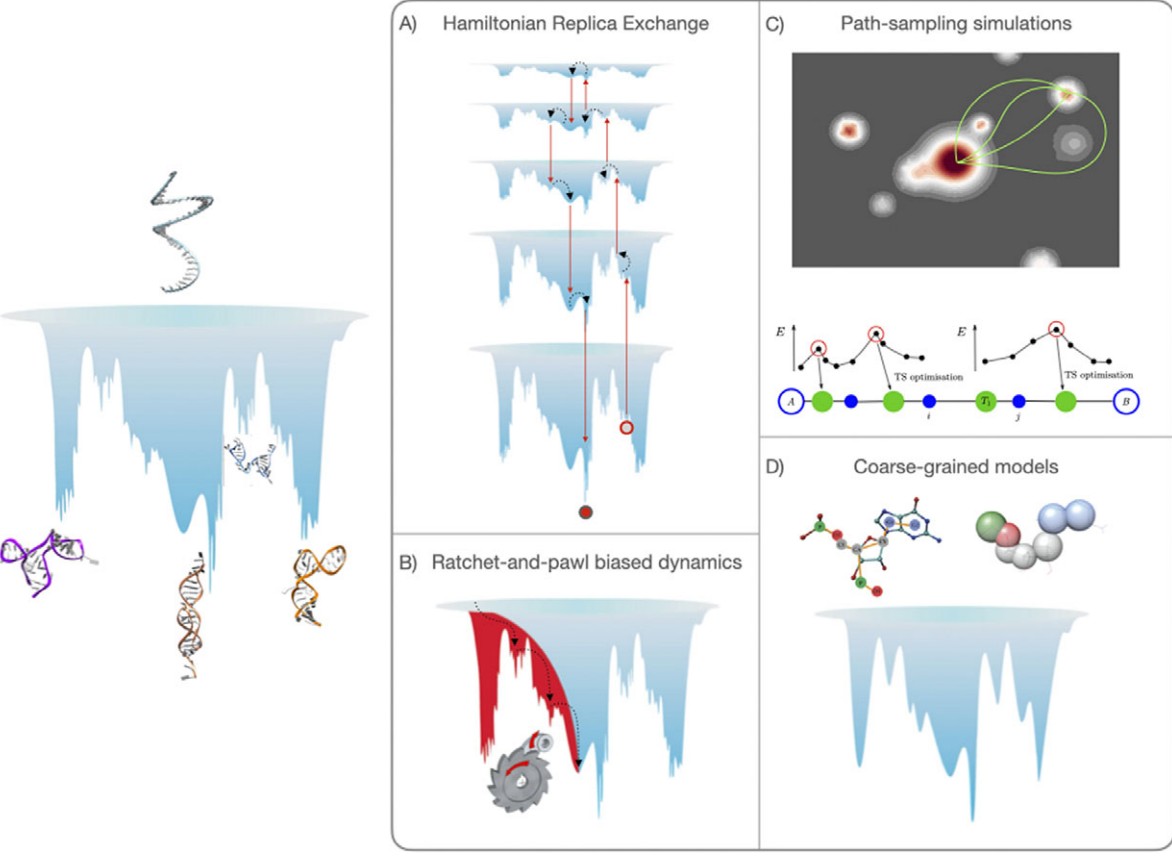

**Fig. 1.** Left: Illustration of how the energy landscape of a polymorphic RNA might look like (the vertical axis represents the energy or free energy of the system). At the top of the landscape, we find high energy unfolded conformations, while we note several deep minima, separated by high barriers, all corresponding to substantially different structures of the molecule. One of these minima might be observed experimentally and referred to as the 'native structure'. Right panel (*a–d*): illustration of how each of the method presented samples the landscape.

resolution models, the computational costs remain high. This shortcoming may be overcome by focusing on a specific region of the system under investigation, reducing the size of the perturbed region, and thus the number of required replicas.

### rMD and the BF approach

rMD simulations are based on introducing a soft history-dependent biasing force to enhance the generation of productive folding trajectories towards a given target structure (Paci and Karplus, 1999). In practice, once a target structure is known experimentally, it is possible to extract some features characteristics of its configuration and define a CV that can be used to guide unfolded structures towards it in a biased MD simulation. In the literature, many CVs exist for biomolecules, ranging from a simple atomic distance or dihedral angle to the radius of gyration, the Root-Mean-Square Deviation (RMSD) and many more depending on the specific feature relevant for the fold of the molecule (Fiorini *et al.*, 2013). The system is free to explore the EL, as long as it follows broadly this predetermined CV, which is a proxy for the reaction coordinate. An external biasing force is switched on when the system backtracks with respect to the CV (see Fig. 1*b*). In RNA and protein folding simulations, one choice for the predetermined CV is obtained from the overlap of the instantaneous and the target atomistic contact map (Camilloni *et al.*, 2011). This approach produces folding trajectories efficiently but requires structural information about the target.

In the ideal case in which CV coincides with the reaction coordinate [the committor function (Ee and Vanden-Eijnden, 2010)], rMD trajectories sample the correct region of configuration space (Cameron and Vanden-Eijnden, 2014; Bartolucci *et al.*, 2018). However, the choice of CV used in RNA folding simulations is only a proxy of the ideal reaction coordinate. Therefore, with rMD it is only possible to obtain an approximate reconstruction of the folding EL. Systematic errors from the biasing force can be minimised by applying the BF filtering procedure (Beccara *et al.*, 2015). In this approach, a variational principle derived from the path integral representation of Langevin dynamics (Onsager and Machlup, 1951) is used to select the folding trajectories generated by rMD that have the highest probability of occurring in the absence of any biasing force.

Apart from the requirement to use structural information about the folded structure, another drawback of rMD simulations is that the generated trajectories only explore part of the EL, namely the region most likely traversed by productive pathways towards the predetermined target structure. While this approach greatly enhances computational efficiency, it prevents the method from exploring other parts of the landscape that may be associated with kinetic trapping.

### DPS for RNA

H-REX and rMD simulations compute trajectories of molecules moving on the EL. DPS (Wales, 2002, 2004) focuses on the

topography of the EL. The EL is considered coarse grained, where only the local minima and transition states that connect them are used as representation. Each transition state connects two local minima, and between any pair of minima, we can identify a discrete path consisting of a series of minima connected by transition states. This representation results in a kinetic transition network, which can then be analysed to obtain kinetic and thermodynamic characteristics, including the associated structures and transition mechanisms.

Through this approach, the topography of the EL is obtained, and this information allows readily for interpretation of mutational data (Röder *et al.*, 2020). As local minima and transition states are well-defined geometrically, they can be located by geometry optimisation, overcoming the dependence on long time scales other simulations suffer from. A shortcoming of the method is the use of implicit solvent representations, which introduces a source of error (Šponer *et al.*, 2018). While it is theoretically possible to use explicit solvent, the increased computational cost currently prevents such set-ups. While free energies can be readily obtained, explorations of higher entropy configurations are difficult. As such, structural transitions between folded structures are generally well resolved, while unfolding events are not. More information and details on how the ELs are explored with DPS can be found in various reviews (Joseph *et al.*, 2017; Röder *et al.*, 2019).

While DPS is most efficient when folded structures are known, the methodology can locate unknown folded structures and new funnels, as demonstrated in the exploration of mutational changes, for example, in 7SK RNA (Röder *et al.*, 2020). However, currently there is no algorithm to guarantee the location of all structures. A useful way around this limitation is to create several possible alternative structures and connect them. Importantly, this approach does not require the structures to be optimised as long as key interactions, such as base pairs are formed.

### Coarse-grained RNA representations

By grouping several atoms into larger particles (grains), the computational exploration of the EL is aided in two ways. Firstly, the CG smooths the EL (see Fig. 1*d*), which removes kinetic traps for the exploration. Secondly, the number of degrees of freedom is reduced, making the computations more tractable. The choice of the mapping between atoms and grains depends on the level of details required and on the kind of interactions that are considered relevant [see (Li and Chen, 2021) for a recent review on the different existing RNA coarse-grained models]. For RNA structures, key elements are base pairing, stacking and electrostatic interactions. In the HiRE-RNA model (High-Resolution Energy model for RNA) (Pasquali and Derreumaux, 2010; Cragnolini *et al.*, 2015), we have chosen to preserve a relatively high resolution with each nucleotide described by 6 or 7 beads. This level of detail, while significantly reducing the number of particles, allows the definition of planes for the nucleobases, reflecting the aromatic rings stacking, and distinguishes different edges of the bases to account for both canonical and non-canonical pairings. While using an implicit solvent, long-range electrostatic effects are accounted for by a Debye–Hückel potential energy term dependent on experimental ionic concentrations in solution. While the development of this coarse-grained model is still on-going, its usefulness for small systems (Stadlbauer *et al.*, 2016; Cragnolini *et al.*, 2017) and when coupled to experimental data (Pasquali *et al.*, 2019; Mazzanti *et al.*, 2021) has been demonstrated.

The obvious shortcoming of any CG methodology is the loss of detail, due to the lower model resolution. In addition, the implicit nature of solvent and ions will impact the observed features. These drawbacks mean the entropy is not faithfully produced within coarse-grained simulations. However, the reduced complexity will allow the study of larger systems and larger scale rearrangements, providing otherwise inaccessible insights.

Despite the fewer degrees of freedom, our coarse-grained MD simulations can still be expensive, with several days of CPU needed to achieve folding of a small molecule of 20–30 nucleotides, although we will be able to achieve much greater speed once the force field will be ported to parallel MD computing.

## A small showcase

In this section, we discuss a few illustrative applications of these methods, emphasising their complementary nature.

### Folding pathway of the human telomerase H-pseudoknot triple helix

This example is a 47-nucleotide RNA, exhibiting an H-pseudoknot (two-interlacing strands) further stabilised by a triple helix (PDB ID 2K96). The system has been studied extensively experimentally (Theimer *et al.*, 2005; Gavory *et al.*, 2006; Kim *et al.*, 2008) and has become a benchmark for modelling (Cho *et al.*, 2009; Biyun *et al.*, 2011; Denesyuk and Thirumalai, 2011), as it contains a pseudoknot, a challenging structural feature and non-canonical interactions leading to triplet formation in the triple helix. This system was also used as test case for the HiRE-RNA model (Cragnolini *et al.*, 2015), and, more recently, to validate the application of variationally optimised rMD to RNAs (Lazzeri *et al.*, 2022). Folding simulations were performed in both instances starting from fully unfolded conformations.

The coarse-grained simulations consisted of a long run with the HiRE-RNA model and replica exchange MD simulations at 64 different temperatures. rMD folding simulations consisted of 100 short runs (each lasting a nominal time interval of 5 ns) with the AMBER ff99 with the Barcelona $\alpha/\gamma$ backbone modification (Perez *et al.*, 2007) and the $\chi$ modification (Zgarbová *et al.*, 2011). It should be emphasised that the simulation time does not directly correlate with the physical transition path time, as the history-dependent bias breaks microscopic reversibility and alters the kinetics. CG simulations required 2 weeks of computation on local cluster in 2015 to achieve the native structure for the first time. rMD simulations required roughly a week of simulation to generate all trajectories on a GPU cluster in 2022. The results of the two simulations are shown in Fig. 2.

The HiRE-RNA simulations yielded the correct native state and identified a sensible folding pathway. Moreover, alternative states were observed, which constitute kinetic traps and are characterised by the formation of non-native secondary structures, leading to alternative folds. These results were in qualitative agreement with experimental evidence of the formation of metastable states and folding intermediates (Kim *et al.*, 2008). Despite the CG, these simulations were computationally expensive and the sampling was not optimal. In particular, it was not possible to give a full assessment of the relative populations of the observed states. The rMD simulations produced more insight into the productive folding pathway to the experimentally observed target structure. It was possible to collect statistically

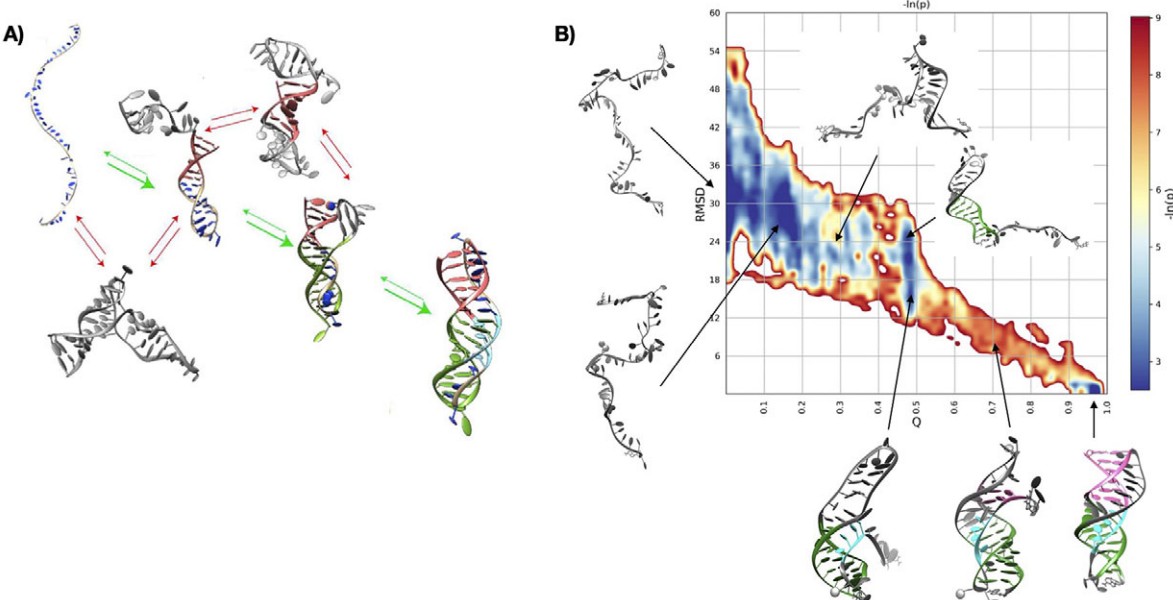

**Fig. 2.** Folding of the human telomerase triple helix performed with unbiased coarse-grained simulations (REMD), allowing to widely explore alternative conformations (*a*) and with biased atomistic simulations allowing to explore the details of intermediate states (*b*).

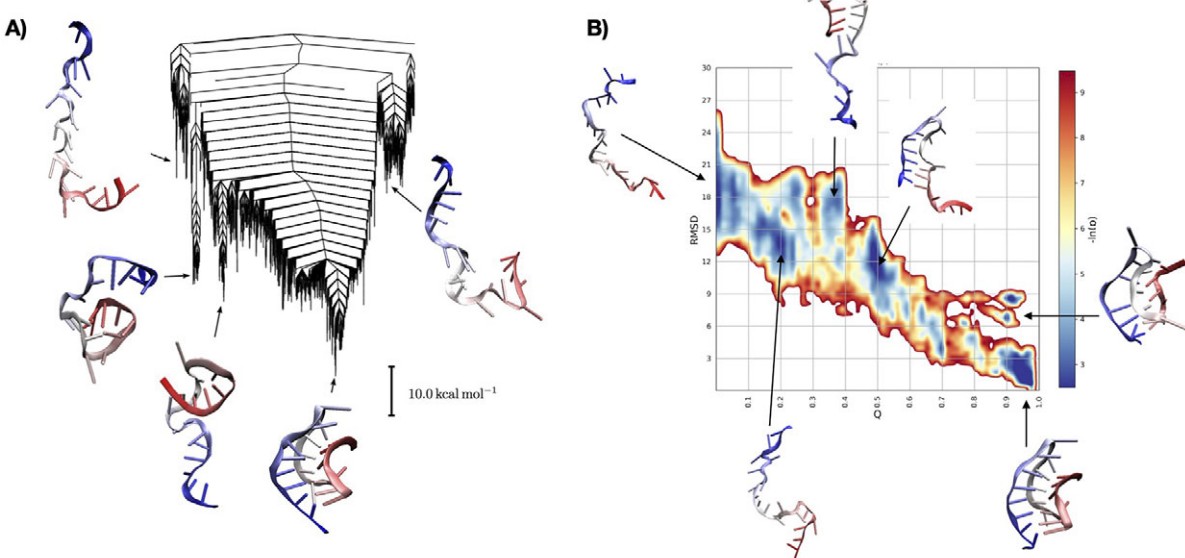

**Fig. 3.** Folding of the small H-pseudoknot PK1 studied with path sampling to obtain its energy landscape (*a*) and with biased folding simulations to study the native folding mechanism (*b*).

relevant populations for the different conformations and generate a heatmap illustrating the folding in terms of formation of native contacts and RMSD with respect to target (see Fig. 2*b*). The results highlighted the ruggedness of the folding landscape, characterised by a multitude of intermediate states. Moreover, we were able to infer the existence of a pronounced bottleneck towards the final stage of folding, when the formation of the pseudoknot takes place. It was also possible to characterise the order of the events of folding in terms of formation of stems and loops and the main path found by our simulations corresponded well with the experimental evidence from thermodynamic studies (Kim *et al.*, 2008) and by simulations by other groups (Cho *et al.*, 2009).

Importantly, both methods lead to the correct folding path and folding intermediates, giving credibility to both methods. In addition, the unbiased CG simulations also identified alternative structures, which are not on the folding pathway. The rMD simulations can provide statistics of the explored states and detailed insight into the interactions along the folding pathway.

### EL and folding pathway of a small H-pseudoknot

For the 22-nucleotide long tmRNA pseudoknot taken from Aquifex aeolicus (PDB ID 2G1W) (Nonin-Lecomte *et al.*, 2006), we performed both a full exploration of the EL with DPS and an exploration of the folding pathway with rMD (see Fig. 3). Both sets of

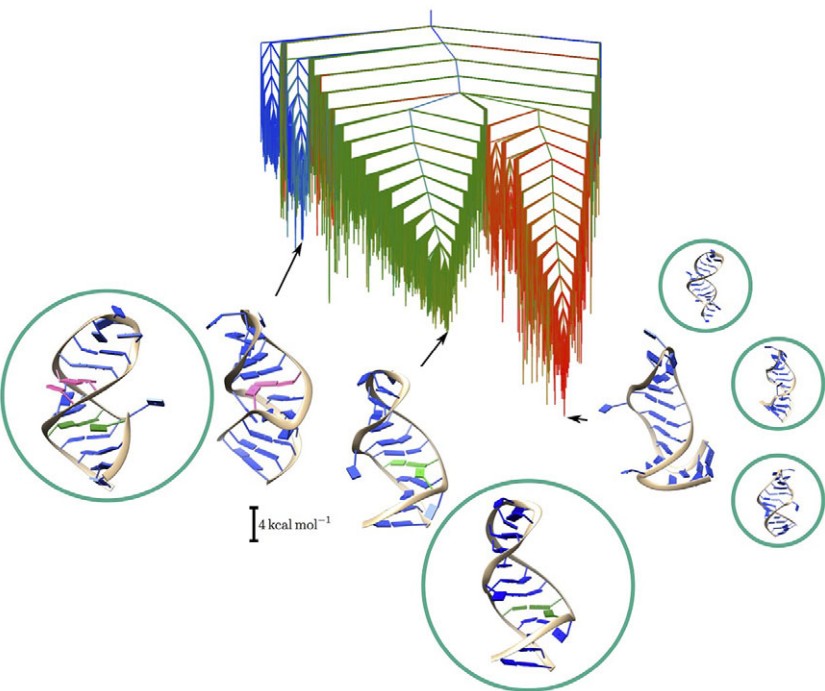

**Fig. 4.** Energy landscape obtained from DPS for the 7SK RNA HP1 hairpin with key structures shown. The structural polymorphism is clearly observable, with three main funnels corresponding to more compact stem loops as observed in X-ray crystallography (blue and green), and more extended structures as observed by NMR experiments (red). Encircled structures correspond to the main clusters observed in H-REX simulations.

simulations used the atomistic AMBER ff99 force field with the Barcelona $\alpha/\gamma$ backbone modification and the $\chi$ modification (Zgarbová *et al.*, 2011). The EL exploration used an implicit solvent model, while the rMD simulations were in explicit solvent.

As the system is small (in fact, it is the smallest known pseudoknot), it was possible to exhaustively explore the EL (Ma *et al.*, 2021). From these simulations, we can again appreciate the presence of a rugged folding funnel (see Fig. 3*a*). The EL is characterised by one main funnel anchored by the native, experimentally observed structure. Some smaller subfunnels exist on the EL, but only small barriers separate them from the main funnel. When analysing the ensemble of structures corresponding to these subfunnels, we detect partially folded states, but no states with alternative secondary structure competing with the native fold. The rMD simulations provided insight into the folding mechanism. As in the previous case, the folding pathways cross a multitude of metastable intermediate states (see Fig. 3*b*).

When integrating the information obtained from these simulation methods, we observe that this RNA has an easily accessible native state, as suggested by the presence of only one major funnel from path sampling and by the absence of significant bottlenecks in the folding simulations. Nonetheless, the folding pathway is not smooth, but the subsequent formation of additional secondary structure elements results in kinetic trapping, rendering the folding process bumpy.

The two approaches support each other in this conclusion. While DPS provides a complete view of the EL, the interaction with the solvent and the high entropy regions are not fully resolved, and rMD provided the missing details for the folding pathway.

### *Exploring polymorphism: 7SK RNA and KSHV's ORF50 transcript*

7SK RNA is a non-coding RNA and part of a ribonucleoprotein complex, which is crucial to transcription regulation by RNA

polymerase II (Wassarman and Steitz, 1991). Its 5′ hairpin (HP1) was characterised experimentally by different methods including X-ray crystallography (Martinez-Zapien *et al.*, 2017), Nuclear magnetic resonance spectroscopy (NMR) (Bourbigot *et al.*, 2016), Small-Angle X-ray Scattering (SAXS) (Brillet *et al.*, 2020) and chemical probing (Lebars *et al.*, 2010; Olson *et al.*, 2022). As a perfect example of RNA polymorphism, the high-resolution methods (NMR and X-ray) detected substantially different structures for this hairpin, including two distinct structures within the same crystal.

The three alternative structures are characterised by a reorganisation of base pairing in the upper portion of the stem. The NMR structure is a hairpin with bulges and only canonical base pairing, while the X-ray structures exhibit non-canonical pairings and some triplets, organised differently in the two structures. The hairpin is the binding site for an affector protein (Egloff *et al.*, 2018), which is crucial to the important biological function of RNA 7SK (Nguyen *et al.*, 2001; Yang *et al.*, 2001). Hence, understanding this structural polymorphism and its implication is of significant importance.

We used DPS and H-REX simulations to study the upper portion of HP1 (27 nucleotides) and further studied a set of mutations (Röder *et al.*, 2020), reported to affect the binding affinity of HP1 (Martinez-Zapien *et al.*, 2017). DPS was initiated from the experimentally observed crystal structures and a multifunnel EL was obtained from sampling, including descriptions of the relative stability and interconversion pathways (see Fig. 4).[2]

The EL revealed the polymorphic character of the HP1 hairpin, and based on the observed structures and their relative energies, we formulated a hypothesis relating the lowest energy X-ray hairpin

---

[2]A detailed description of how this computational study was conducted is reported in Röder and Pasquali (2021).

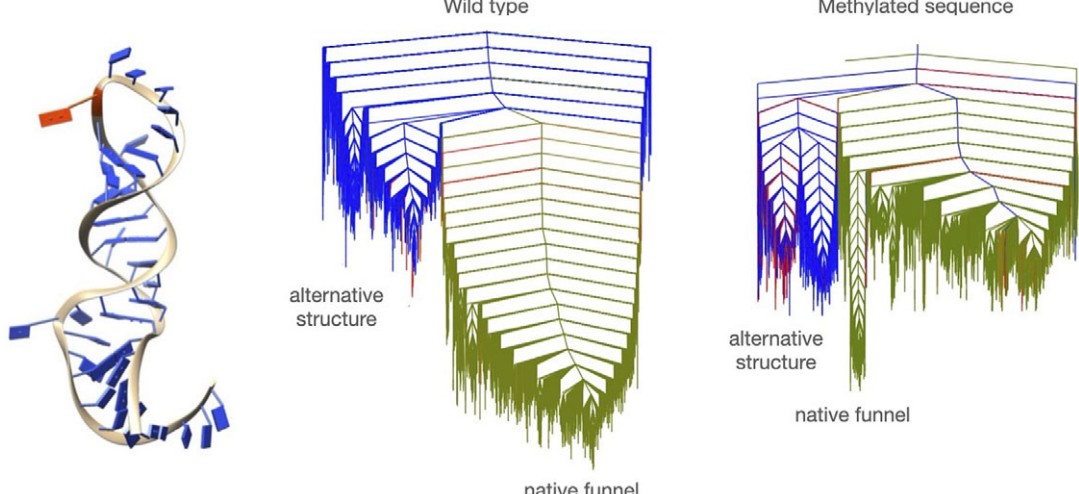

**Fig. 5.** EL of wild type and methylated sequence for ORF50. One of the lowest energies structures is shown with the site of methylation highlighted in red.

structure to the binding of the affector protein. From our exploration of the ELs for various mutants, we were able to draw a correlation between specific mutations and protein binding affinity, providing a mechanical explanation of the observed mutational effects.

Two sets of H-REX simulations complemented the EL exploration with DPS, each starting from one of the observed crystal structures. Simulations were performed for 100 ns using 20 replicas, with a Hamiltonian coupling $\lambda$ ranging from 1 to 0. Due to the large size of the solvated system (roughly 30,000 atoms), convergence for such simulations is difficult to achieve. Nonetheless, the findings from the extended trajectories agree with the observations from path sampling. The clusters of structures corresponded well to the structures found in the main funnels of the EL. The agreement between the simulations enabled us to exclude a prominent role of structural water molecules or ions, which is not possible from the implicit solvent representation used in DPS. While DPS did not necessarily explore the high-energy portions of the landscape, which would require unfolding and potential refolding into different structures, the H-REX simulations did explore these regions. We would therefore expect H-REX to be able to depart more significantly from the initial structures and possibly find new structures with a full reorganisation of the molecule. In our simulations, however, we only located states already explored by DPS. The experimental evidence combined with the exploration of two different kind of simulations gives us some confidence that we have probably explored all the biologically relevant structures of the system.

Another area of growing interest is the study of post-translational modifications. In a recent study, we investigated a post-translational methylation of an RNA hairpin (Röder *et al.*, 2022). As many RNAs are subject to methylation (Zaccara *et al.*, 2019), it is important to understand how this modification impacts the adopted structures and how the equilibrium between possible alternative structures is altered. In the case of the RNA transcript of open reading frame 50 (ORF50) of Kaposi's sarcoma-associated herpes virus, which encodes the replication and transcription activator protein required for viral activation (Guito and Lukac, 2012), methylation stabilises the RNA transcript, leading to effective viral replication (Baquero-Perez *et al.*, 2019). Here, DPS was used based

on experimentally observed secondary structures (Baquero-Perez *et al.*, 2019) only.

Our study revealed the existence of several structural basins, with the native structure occupying the lowest energy states. A set of higher energy structures, which allow interactions of the transcript with proteins (in this case, an m6A reader) is found as well. In the unmethylated system, these structures are effectively inaccessible, while the methylation reduces the energy difference significantly, leading higher occupation of these states, which can recruit the m6A reader (see Fig. 5).

These results highlight the importance of studies of mutations and chemical modifications, as the unmodified sequence might exhibit polymorphism, but it cannot be detected in experiment, while small alterations lead to significant changes in the EL, which may lead to detectable polymorphism.

## Conclusions and perspectives

From these case studies, we can draw some general conclusions on the specificity of RNA folding and on what can constitute a profitable strategy to tackle larger and more complex systems. For all systems we have studied, even the simplest, we find a rugged EL, i.e. it is characterised by the presence of many locally stabilised structures, corresponding to the subsequent formation of local secondary structures. These configurations may be part of a single funnel, most likely for small systems, or, if they exhibit significant difference in their secondary structure, belong to competing funnels, a feature likely observed for larger RNA molecules.

A successful strategy to investigate the possible structures is the combination of several simulation methods. Secondary structure predictions based on bioinformatics may already reveal some of the complexity of the RNA folding problem. If a single dominant structure emerges, it is likely that the EL is less complex and exhibits a single main funnel. In such cases, it is likely that canonical base pairs are adopted, and chemical probing may give further confidence in such predictions. Non-canonical interactions may be important to local structural details. Of course, in these scenarios, complexity may arise from additional effects, such as post-translational modifications.

If, on the other hand, multiple secondary structures are proposed, a multiscale approach can be useful. Using a coarse-grained

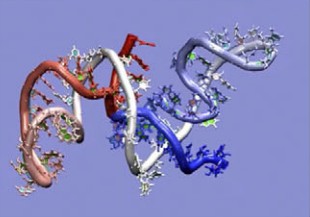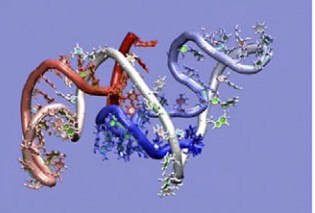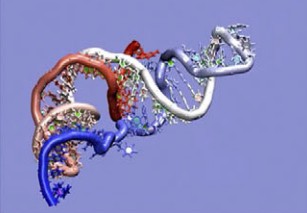

**Fig. 6.** Sample structures on the conversion pathway between the native state of the frameshifting pseudoknot of SARS-CoV-2 (left) and an alternative structure with no pseudoknot (right).

model, initial scouting of the EL can yield a survey of possible alternative structures, leading to an identification of the main folding funnels. Subsequent all atom simulations can then be used to investigate details on the EL, seeded by the structures obtained from the coarse-grained simulations.

With this approach in mind, we recently started investigating the frameshifting pseudoknot of SARS-CoV-2, for which a structure is known experimentally but for which both experimental and simulation data suggest the existence of alternative structures (Schlick *et al.*, 2021; Yan *et al.*, 2022). We first simulated the system at a relatively high temperature with the CG model to generate seeds for DPS in order to speed up the EL exploration. Then, using DPS, a search for the conversion path between the native structure and a proposed structure lacking the characteristic pseudoknot was initiated with these seeds. Given the size of the system, simulations are computationally very demanding and still ongoing, but preliminary results show the possible conversion path between the two states (Fig. 6).

Another research direction is based on the extensive conformational sampling and access to free- EL explorations provided by H-REX, which could offer decisive insights into key phenomena related to the RNA World hypothesis and the origins of life, as currently studied by some of us. Previous studies on protein enzyme systems have shown the crucial importance of proper conformational sampling of the reactant and product states to understand the chemical reactivity of these biological objects (*Maffucci et al., 2020a*). We thus aim at understanding how a ribozyme's accessible conformations can affect its reactivity. Secondly, the end product of template-based RNA replication in abiotic conditions is an RNA dimer. However, these are known to be very stable constructs, with high denaturation temperatures. Inspired by the use of the REST2 strategy for the study of protein melting properties (Stirnemann and Sterpone, 2015, 2017; Maffuci et al. 2020*b*), we are currently trying to understand how RNA duplexes separate upon temperature increase, and how this depends on the strand sequence.

While many computational methods exist to study biomolecules, the challenges encountered by the complexity of RNA folding means that the best strategy for computational studies to yield useful biological insight, rests on the combination of multiple approaches to overcome individual shortcomings and access time and size scales otherwise inaccessible. While data-based structural predictions are important, they require the additional insight from physical modelling, especially to understanding the dynamic, polymorphic nature of RNA. Simulations of RNA have been used for decades, but the maturity of many methods and the growing understanding of RNA mean that a new chapter of research has opened up, where computational approaches, if used properly, will routinely provide exciting insights into the nature of RNA.

**Open peer review.** To view the open peer review materials for this article, please visit http://doi.org/10.1017/qrd.2022.19.

**Data availability statement.** Simulation data are available through links relative to the original articles presenting the material.

**Author contributions.** K.R., G.S., P.F. and S.P. all contributed to the developments of the various methods presented in the manuscript and all contributed to its writing. S.P. conceived the article and made the figures.

**Financial support.** K.R. is funded by the Cambridge Philosophical Society. K.R. and S.P. thank Université Paris Cité for a visiting fellowship for K.R. Some of the research mentioned here has received funding from the European Research Council under the European Union's Eighth Framework Program (H2020/20142020)/ERC Grant Agreement No. 757111 (G.S.). This work was also supported by the 'Initiative d'Excellence' program from the French State (Grant 'DYNAMO', ANR-11-LABX-0011-01 to GS). S.P. is thankful for the computing time allocated to the French national computing grant A0090710584. This work was partially supported by a STSM Grant from COST Action CA17139 (eutopia.unitn.eu) funded by COST (www.cost.eu).

**Conflict of interest.** K.R., G.S. and S.P. do not have any conflict of interest. P.F. is co-founder and shareholder of Sibylla Biotech SPA, a company exploiting molecular simulations to perform early-stage drug discovery.

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
