## [Reviewer Report]

*Comments to Author*: Here the authors describe several computational methods commonly used to study RNA. They discuss Hamiltonian-replica exchange MD, ratchet-and-pawl MD, discrete path sampling and HiRE-RNA coarse-graining scheme. Next they discuss several case studies where these methods have been used. Finally they report the limitations of these methods and how multiple methods will better help understand experimental results on RNA polymorphism.

This reviewer finds well written but however has few concerns and confusion regarding the author’s manuscript as listed below.

1. The title is not very informative, and the reviewer suggests the author search for an informative title. Especially “methods for the greater good” add very little meaning to the title.

2. In the introduction, the sentence “RNA folding was thought to be an easier problem to solve than protein folding.” requires a citation at the end of the statement.

3. In the introduction the sentence “Moreover, large RNA architectures often present pseudoknots (non-nested arrangements of base pairs), which can not yet be treated efficiently by traditional secondary structure prediction algorithms and that are only now becoming accessible in combined strategies using machine learning (Sato and Kato 2022; Wang, Liu, et al. 2019).” is quite large and the reviewer requests the author consider breaking this sentence down.

4. In the introduction, this sentence requires at least two or more citation “The multiple hydrogen bond donor and acceptor sites of the nucleobases allow for a multitude of base pairs, which have been reported experimentally” in favor this claim.

5. In the introduction, for the line “The small alphabet combined with the multitude of possible base pairs results in RNA folding being as or even more complex than protein folding.”, the authors does not explain what small alphabet mean. Furthermore, they claim RNA folding being just a complex or more than protein folding however this sentence does not have a citation in it’s favor.

6. In the introduction, the authors do not explain what “minimal frustration” is. This needs to be defined as well mention what it means for a structure to have a higher frustration vs lower frustration. Without this definition the rest of their argument is hard to understand. Furthermore, while on the subject of protein folding a lot more recent papers have come out from the David Baker group (RoseTTAFold) and from DeepMind (AlphaFold2). The reviewer suggest the author explain computational protein folding those perspective.

7. In the introduction, the sentence “The energy landscape contains the all information about the structures, their energy, and interconversion pathways between them.” seem to have been constructed strangely and the reviewer ask modification to this sentence. In the same paragraph, the reviewer is not quite sure what the authors mean by “Any experiment or simulation probes the energy landscape, though various methods do so in different ways, which can be explicit or implicit.” and asks the author to clarify this sentence in their text.

8. In the introduction, the paragraph with the line “broken ergodicity exhibited biomolecular energy landscapes” is lacking details and the reviewer fails to see the point of this paragraph and how it connects to the next paragraph. Therefore, the reviewer request the author to rewrite this paragraph such that it can relate to why MD simultations for RNA are difficult.

In summary the reviewer feels that the introduction is not strongly connected to the prespective that is being highlighted here and request the authors

9. In the replica exchange section the opening sentence “As these simulations do not require the definition of collective variables (colvars), the sampling is unbiased.” is not introduced well and reviewer request this sentence be restructured. For example up until now, nothing is mentioned about colvars. Additionally, the reviewer is also confused how the sampling is unbiased if the potential energy is tempered with.

10. This sentence “Importantly, entropic contributions for partially folded structures are captured in this approach” requires a citation.

11. In the paragraphs about REST2 scheme, the authors explain the approach and gives with proteins of different sizes. Has there been no studies done with RNA? If yes, then the authors should also briefly explain that. If they explain it later in the manuscript they should mention that too. Else, the authors should mention that REST2 scheme has not been applied for studying RNA.

12. By the time the reviewer reached section “Ratchet-and-pawl molecular dynamics and the Bias Functional approach”, the reiviewer has already forgotten what rMD stands for and had to scroll up to figure it out. The point, the reviewer is trying to make here is that it might be nice to re-introduce the abbreviation at the beginning of this section.

13. The opening sentence “rMD simulations are based on introducing a soft history-dependent biasing force to enhance the generation of productive folding trajectories towards a given target structure” of the rMD section is not explained well and the reviewer ask the authors to explain this clearly. Additionally this sentence is not cited.

14. Because the authors previously did not explain colvars mean for biological system such as proteins and RNA, their explanation using colvars also suffers in rMD section.

15. This sentence “In RNA and protein folding simulations, one choice for the predetermined CV is obtained from the overlap of the instantaneous and the target atomistic contact map.” requires a citation.

16. In the sentence “Therefore, with rMD it is only possible to obtain an approximate reconstruction of the folding EL” what is EL?

17. Langevin dynamics needs to be cited.

18. The shorthand “H-REX” has not been previously introduced or at the beginning of the section “Discrete pathsampling for RNA”

19. This sentence “The energy landscape is coarse-grained into a set of local minima and transition states that connect them.” is not explained well.

20. There are grammatical issues with this sentence “we can identified a discrete path consisting of a series of minima connected by transition states.”

21. In the section Discrete pathsampling of RNA and for the sentence “which introduces a source of error.”, the authors didn’t give examples what source of error can cause to the simulation. Additionally this sentence needs citations in case the readers wanted to know more about.

22. This sentence “More information and details on how the energy landscapes are explored can be found in various reviews” does not have a basis. The section is about Discrete pathsampling of RNA, however the last sentence of this section is very general. The reviewer suggest the authors to stay true to the section and come up with a better closing sentence.

23. Could the authors point out what does “HiRE” in HiRE-RNA stand for?

24. In the last paragraph of the section “Folding pathway of the human telomerase H-pseudoknot triple helix”, could the authors comment on how long it took (computational (CPU/GPU) hours) for unbiased CG simulations and the rMD simulations in this particular case and additionally provide further guidance for readers for what sizes of systems can these simulations be used complementarily and beyond which it becomes impractical?

25. The sentence “The hairpin is the binding site for an affector protein[cite], which is crucial to the important biological function of RNA 7SK[cite, …]” requires at least two or more citations as mentioned in square-brackets in the sentence.

---

## [Reviewer Report]

*Comments to Author*: It is an interesting overview of folding modelling literature of small(er) RNAs, focusing on the recent studies published by the authors. The paper is generally well written, the overall concept is smart, it has a good logical structure, and should be publishable. However, a major revision is suggested, as outlined below, as one of the parts is not in an acceptable form.

Major criticism

The H-rex subchapter on pp. 3-4 is not realistic, specifically with its emphasis on the REST2 method. This part lacks any comments on the limitation of the methods. In addition, while the paper should be about RNA simulations, the paragraph is presently supported only by citations to protein or enzyme works which in addition appear to be self-citations by one of the authors. I have some doubts if proteins with 100-200 residues were convergently sampled by REST2. Nevertheless, unfortunately, a complete failure of the REST2 method for even the shortest RNA 8-mer tetraloops has been recently demonstrated on an aggregate millisecond time scale, 100+ microseconds per replica; no sign of a convergence (Mlynsky et al., JOURNAL OF CHEMICAL THEORY AND COMPUTATION 18 (4), pp.2642-2656). There is clear lack of decorrelation in the replica ladder which even deepens once there is some folding event in the ladder. The folded continuous trajectories appeared to be pinned down at the bottom of the ladder. REST2 sampling might even deteriorate compared to plain simulations. Deeper theoretical roots of this can be found in Zuckerman et al., https://www.annualreviews.org/doi/abs/10.1146/annurev-biophys-042910-155255. So, it is a problem which should be known in the field but is often ignored. However, even for proteins the performance of REST2 has been very seriously questioned, Appadurai et al. https://www.nature.com/articles/s41467-021-21105-7 In this work other potential issues of REST2 are mentioned, affecting its sampling efficacy, such as cold solvent slowing down the conformational diffusion (analogous problem was known in the past to affect the Locally Enhanced Sampling approach in explicit solvent). In summary, the HREX paragraph is not in an acceptable form and should be rewritten completely, focused on the RNA RE works (such as the above paper, Bergonzo et al., RNA 21 (9), pp.1578-1590 and several others in the literature; some literature search would be useful) and give a fair assessment of the limitations. I think the above-noted principal problems of REST2 may be responsible for some of the problematic sampling cases hinted in the other parts of the paper. Based on the recent works REST2 is most likely not very suitable method for RNA sampling, unless combined with some CV-class approaches. Even that can help only for specific systems where dimensionality reduction is possible by CVs.

On the other hand the problematic efficiency of the explicit solvent atomistic methods highlights the need for the coarse-grained approaches, not mentioning the force-field limitations, making the story logical.

Minor comments.

In the DPS part of the paper, one point should be clarified for the reader. How much does the method depend on availability of the folded structures? I tend to think that the method is still not able to generally find unknown folds, if they are different and separated by substantial barriers from the funnel under investigation, i.e., I would assume there easily could be entirely missed parts of the landscape if prior information is not available. Could the authors briefly comment to it? Otherwise I agree it is a useful method to see the landscape structure, though there is the implicit solvent limitation (which is nevertheless clearly articulated in the paper, similarly to some other limitations). I assume the method can analyze roughness of a known funnel but I doubt it can find new funnels in case of major kinetic partitioning. Is that correct?

The paper, albeit declared to be about RNA, is making several excursions to DNA quadruplexes (GQs). However, it may be somewhat excessive and even misleading. DNA GQs are a specific case of biopolymers where the kinetic partitioning originates from the interrelation between the folds and the syn-anti orientations of guanines. It is however not relevant to RNA GQs, as RNA GQs are all-anti and thus always all-parallel. The main source of frustration of the folding landscape in DNA GQs is thus not relevant for RNA GQs. Most recently it was reviewed in Sponer et al. https://www.sciencedirect.com/science/article/abs/pii/S0065774320300154, experimental view Grun&Schwalbe, Biopolymers 2021 DOI: 10.1002/bip.23477 As presently explained it is misleading or incomplete; the picture from DNA GQs cannot be extrapolated to RNA GQs and in fact, as far as I know, it does not have analogy in RNA folding. It is unique to DNA G-rich single strands due to the possibility of syn-anti partitioning which leads to numerous competing basins of attraction with different syn-anti patterns.

In the Introductions the authors might note that RNAs massively interact with proteins, so they often co-fold with proteins and are re-modelled. And the hierarchical nature of RNA structure makes the RNA structure extremely complicated for predictions from the sequences, nice visualization of this problem is in Fig. 2 of Jaeger et al. NUCLEIC ACIDS RESEARCH 37 (1), pp.215-230. Proteins can be better predicted if they have domain folding.

Another minor point for consideration.

The text emphasizes the concept of native state which reads as synonym to a well folded structure. However, I think the authors might note disorder (many RNAs may be disordered, just they are underrepresented in contemporary research due to lack of methods). In addition, one can think even about disordered state as being the native state, if it is the biochemically relevant ensemble. The intro is correct but somewhat too much based on traditional concepts reflecting the world of simple fast-folding proteins that have frustration-minimizing native basin and funnel folding.

It appears that the Journal is imposing some quite strict requirements on the length of the paper, which may be responsible for some of the weaknesses of the present version noted above. I strongly recommend that the Editor gives the authors some flexibility regarding the length, if needed. MD technique is very complex field with countless methodological approaches, and very tricky reliability issues. MD works should be interpreted upon considering their limitations. It is in the interest of the readers, but ultimately also of the authors and the Journal. Otherwise the message could be confusing. (Myself I would not be capable to write it within such length limits).

---

## [Reviewer Report]

*Comments to Author*: Reviewer #1: It is an interesting overview of folding modelling literature of small(er) RNAs, focusing on the recent studies published by the authors. The paper is generally well written, the overall concept is smart, it has a good logical structure, and should be publishable. However, a major revision is suggested, as outlined below, as one of the parts is not in an acceptable form.

Major criticism

The H-rex subchapter on pp. 3-4 is not realistic, specifically with its emphasis on the REST2 method. This part lacks any comments on the limitation of the methods. In addition, while the paper should be about RNA simulations, the paragraph is presently supported only by citations to protein or enzyme works which in addition appear to be self-citations by one of the authors. I have some doubts if proteins with 100-200 residues were convergently sampled by REST2. Nevertheless, unfortunately, a complete failure of the REST2 method for even the shortest RNA 8-mer tetraloops has been recently demonstrated on an aggregate millisecond time scale, 100+ microseconds per replica; no sign of a convergence (Mlynsky et al., JOURNAL OF CHEMICAL THEORY AND COMPUTATION 18 (4), pp.2642-2656). There is clear lack of decorrelation in the replica ladder which even deepens once there is some folding event in the ladder. The folded continuous trajectories appeared to be pinned down at the bottom of the ladder. REST2 sampling might even deteriorate compared to plain simulations. Deeper theoretical roots of this can be found in Zuckerman et al., https://www.annualreviews.org/doi/abs/10.1146/annurev-biophys-042910-155255. So, it is a problem which should be known in the field but is often ignored. However, even for proteins the performance of REST2 has been very seriously questioned, Appadurai et al. https://www.nature.com/articles/s41467-021-21105-7 In this work other potential issues of REST2 are mentioned, affecting its sampling efficacy, such as cold solvent slowing down the conformational diffusion (analogous problem was known in the past to affect the Locally Enhanced Sampling approach in explicit solvent). In summary, the HREX paragraph is not in an acceptable form and should be rewritten completely, focused on the RNA RE works (such as the above paper, Bergonzo et al., RNA 21 (9), pp.1578-1590 and several others in the literature; some literature search would be useful) and give a fair assessment of the limitations. I think the above-noted principal problems of REST2 may be responsible for some of the problematic sampling cases hinted in the other parts of the paper. Based on the recent works REST2 is most likely not very suitable method for RNA sampling, unless combined with some CV-class approaches. Even that can help only for specific systems where dimensionality reduction is possible by CVs.

On the other hand the problematic efficiency of the explicit solvent atomistic methods highlights the need for the coarse-grained approaches, not mentioning the force-field limitations, making the story logical.

Minor comments.

In the DPS part of the paper, one point should be clarified for the reader. How much does the method depend on availability of the folded structures? I tend to think that the method is still not able to generally find unknown folds, if they are different and separated by substantial barriers from the funnel under investigation, i.e., I would assume there easily could be entirely missed parts of the landscape if prior information is not available. Could the authors briefly comment to it? Otherwise I agree it is a useful method to see the landscape structure, though there is the implicit solvent limitation (which is nevertheless clearly articulated in the paper, similarly to some other limitations). I assume the method can analyze roughness of a known funnel but I doubt it can find new funnels in case of major kinetic partitioning. Is that correct?

The paper, albeit declared to be about RNA, is making several excursions to DNA quadruplexes (GQs). However, it may be somewhat excessive and even misleading. DNA GQs are a specific case of biopolymers where the kinetic partitioning originates from the interrelation between the folds and the syn-anti orientations of guanines. It is however not relevant to RNA GQs, as RNA GQs are all-anti and thus always all-parallel. The main source of frustration of the folding landscape in DNA GQs is thus not relevant for RNA GQs. Most recently it was reviewed in Sponer et al. https://www.sciencedirect.com/science/article/abs/pii/S0065774320300154, experimental view Grun&Schwalbe, Biopolymers 2021 DOI: 10.1002/bip.23477 As presently explained it is misleading or incomplete; the picture from DNA GQs cannot be extrapolated to RNA GQs and in fact, as far as I know, it does not have analogy in RNA folding. It is unique to DNA G-rich single strands due to the possibility of syn-anti partitioning which leads to numerous competing basins of attraction with different syn-anti patterns.

In the Introductions the authors might note that RNAs massively interact with proteins, so they often co-fold with proteins and are re-modelled. And the hierarchical nature of RNA structure makes the RNA structure extremely complicated for predictions from the sequences, nice visualization of this problem is in Fig. 2 of Jaeger et al. NUCLEIC ACIDS RESEARCH 37 (1), pp.215-230. Proteins can be better predicted if they have domain folding.

Another minor point for consideration.

The text emphasizes the concept of native state which reads as synonym to a well folded structure. However, I think the authors might note disorder (many RNAs may be disordered, just they are underrepresented in contemporary research due to lack of methods). In addition, one can think even about disordered state as being the native state, if it is the biochemically relevant ensemble. The intro is correct but somewhat too much based on traditional concepts reflecting the world of simple fast-folding proteins that have frustration-minimizing native basin and funnel folding.

It appears that the Journal is imposing some quite strict requirements on the length of the paper, which may be responsible for some of the weaknesses of the present version noted above. I strongly recommend that the Editor gives the authors some flexibility regarding the length, if needed. MD technique is very complex field with countless methodological approaches, and very tricky reliability issues. MD works should be interpreted upon considering their limitations. It is in the interest of the readers, but ultimately also of the authors and the Journal. Otherwise the message could be confusing. (Myself I would not be capable to write it within such length limits).

Reviewer #2: Here the authors describe several computational methods commonly used to study RNA. They discuss Hamiltonian-replica exchange MD, ratchet-and-pawl MD, discrete path sampling and HiRE-RNA coarse-graining scheme. Next they discuss several case studies where these methods have been used. Finally they report the limitations of these methods and how multiple methods will better help understand experimental results on RNA polymorphism.

This reviewer finds well written but however has few concerns and confusion regarding the author’s manuscript as listed below.

1. The title is not very informative, and the reviewer suggests the author search for an informative title. Especially “methods for the greater good” add very little meaning to the title.

2. In the introduction, the sentence “RNA folding was thought to be an easier problem to solve than protein folding.” requires a citation at the end of the statement.

3. In the introduction the sentence “Moreover, large RNA architectures often present pseudoknots (non-nested arrangements of base pairs), which can not yet be treated efficiently by traditional secondary structure prediction algorithms and that are only now becoming accessible in combined strategies using machine learning (Sato and Kato 2022; Wang, Liu, et al. 2019).” is quite large and the reviewer requests the author consider breaking this sentence down.

4. In the introduction, this sentence requires at least two or more citation “The multiple hydrogen bond donor and acceptor sites of the nucleobases allow for a multitude of base pairs, which have been reported experimentally” in favor this claim.

5. In the introduction, for the line “The small alphabet combined with the multitude of possible base pairs results in RNA folding being as or even more complex than protein folding.”, the authors does not explain what small alphabet mean. Furthermore, they claim RNA folding being just a complex or more than protein folding however this sentence does not have a citation in it’s favor.

6. In the introduction, the authors do not explain what “minimal frustration” is. This needs to be defined as well mention what it means for a structure to have a higher frustration vs lower frustration. Without this definition the rest of their argument is hard to understand. Furthermore, while on the subject of protein folding a lot more recent papers have come out from the David Baker group (RoseTTAFold) and from DeepMind (AlphaFold2). The reviewer suggest the author explain computational protein folding those perspective.

7. In the introduction, the sentence “The energy landscape contains the all information about the structures, their energy, and interconversion pathways between them.” seem to have been constructed strangely and the reviewer ask modification to this sentence. In the same paragraph, the reviewer is not quite sure what the authors mean by “Any experiment or simulation probes the energy landscape, though various methods do so in different ways, which can be explicit or implicit.” and asks the author to clarify this sentence in their text.

8. In the introduction, the paragraph with the line “broken ergodicity exhibited biomolecular energy landscapes” is lacking details and the reviewer fails to see the point of this paragraph and how it connects to the next paragraph. Therefore, the reviewer request the author to rewrite this paragraph such that it can relate to why MD simultations for RNA are difficult.

In summary the reviewer feels that the introduction is not strongly connected to the prespective that is being highlighted here and request the authors

9. In the replica exchange section the opening sentence “As these simulations do not require the definition of collective variables (colvars), the sampling is unbiased.” is not introduced well and reviewer request this sentence be restructured. For example up until now, nothing is mentioned about colvars. Additionally, the reviewer is also confused how the sampling is unbiased if the potential energy is tempered with.

10. This sentence “Importantly, entropic contributions for partially folded structures are captured in this approach” requires a citation.

11. In the paragraphs about REST2 scheme, the authors explain the approach and gives with proteins of different sizes. Has there been no studies done with RNA? If yes, then the authors should also briefly explain that. If they explain it later in the manuscript they should mention that too. Else, the authors should mention that REST2 scheme has not been applied for studying RNA.

12. By the time the reviewer reached section “Ratchet-and-pawl molecular dynamics and the Bias Functional approach”, the reiviewer has already forgotten what rMD stands for and had to scroll up to figure it out. The point, the reviewer is trying to make here is that it might be nice to re-introduce the abbreviation at the beginning of this section.

13. The opening sentence “rMD simulations are based on introducing a soft history-dependent biasing force to enhance the generation of productive folding trajectories towards a given target structure” of the rMD section is not explained well and the reviewer ask the authors to explain this clearly. Additionally this sentence is not cited.

14. Because the authors previously did not explain colvars mean for biological system such as proteins and RNA, their explanation using colvars also suffers in rMD section.

15. This sentence “In RNA and protein folding simulations, one choice for the predetermined CV is obtained from the overlap of the instantaneous and the target atomistic contact map.” requires a citation.

16. In the sentence “Therefore, with rMD it is only possible to obtain an approximate reconstruction of the folding EL” what is EL?

17. Langevin dynamics needs to be cited.

18. The shorthand “H-REX” has not been previously introduced or at the beginning of the section “Discrete pathsampling for RNA”

19. This sentence “The energy landscape is coarse-grained into a set of local minima and transition states that connect them.” is not explained well.

20. There are grammatical issues with this sentence “we can identified a discrete path consisting of a series of minima connected by transition states.”

21. In the section Discrete pathsampling of RNA and for the sentence “which introduces a source of error.”, the authors didn’t give examples what source of error can cause to the simulation. Additionally this sentence needs citations in case the readers wanted to know more about.

22. This sentence “More information and details on how the energy landscapes are explored can be found in various reviews” does not have a basis. The section is about Discrete pathsampling of RNA, however the last sentence of this section is very general. The reviewer suggest the authors to stay true to the section and come up with a better closing sentence.

23. Could the authors point out what does “HiRE” in HiRE-RNA stand for?

24. In the last paragraph of the section “Folding pathway of the human telomerase H-pseudoknot triple helix”, could the authors comment on how long it took (computational (CPU/GPU) hours) for unbiased CG simulations and the rMD simulations in this particular case and additionally provide further guidance for readers for what sizes of systems can these simulations be used complementarily and beyond which it becomes impractical?

25. The sentence “The hairpin is the binding site for an affector protein[cite], which is crucial to the important biological function of RNA 7SK[cite, …]” requires at least two or more citations as mentioned in square-brackets in the sentence.